# MODEL FOR SEGMENTATION OF FOREST LOGGING IN NORTHERN LATITUDES USING SENTINEL-2 IMAGES BASED ON AN ENSEMBLE OF MULTI-SPECIALIZED SEGMENTERS

**Andrei Melnikov**[1,2]**, Oleg Sokolkov**[1,2]

[1] Ugra Research Institute of Information Technologies, Khanty-Mansiysk, Russia
[2] Ugra State University, Khanty-Mansiysk, Russia
`{melnikovav,sokolkovoi}@uriit.ru`

## ABSTRACT

The problem of automated decryption of illegal logging in the forests of the Khanty-Mansiysk Autonomous Okrug-Yugra is considered. Due to the human factor, a share of such areas is missed. We are talking about the impossibility of catching some of the violators who avoid a fine. The option of involving more operators in the work may naturally entail unreasonable financial costs. By applying methods and algorithms of neural network segmentation, we intend to significantly reduce the costs of decrypting changes in forest lands for violations of environmental legislation. This task was already considered for our district earlier [1, 2]. In this article, we present a solution based on an ensemble of multi-specialized segmentors, which exceeds the mentioned one by 3.1% in F1-score. As an initial data set, we use summer Sentinel-2 satellite images of the Khanty-Mansiysk Autonomous Okrug-Yugra territory for 2018-2022 with manually marked fellings.

## 1 INTRODUCTION

One of the key problems of monitoring the forest cover of the Khanty-Mansiysk Autonomous Okrug - Yugra is its fairly large area, comparable to the area of France. Manual (non-automated) decoding of images by a large group of operators with the required frequency will be extremely expensive. When using a small group - a long and insufficiently effective procedure due to the human factor.

The approach of automatic recognition (i.e. without human participation) by the neural network model was excluded from consideration, since it raises a number of difficult to resolve issues related to legislative aspects. Instead of automation, in order to increase efficiency while maintaining an acceptable level of financial costs, the path of process automation was chosen. The use of such a system can significantly simplify the process by facilitating monotonous work, reducing the load on the operator. In this case, he is supposed to confirm or reject the results in a specially designed software environment.

Using a software solution based on artificial intelligence, forest cover is scanned for disturbances in real time. If the neural network model predicts forest felling in a certain area, this area is displayed to the operator for decision-making. If this is not a false positive, the operator records the fact and transmits the data to the relevant authorities.

## 2 BASELINE

Previously, a model for segmentation of forest fellings for the Khanty-Mansiysk Autonomous Okrug-Yugra was developed [2,3], which was subsequently improved. This solution has reached a practical level. As part of the pilot testing of the solution, it was established that when using software based on the proposed model, the expert's work time is reduced by 4 times (from 6 to 1.5 hours).

The quality of the model was assessed using a series of experiments comparing different neural network architectures. The highest performance was demonstrated by the SegFormer-B5 transformer neural network model [4], which achieved an F1-score of 76.7%. But it is important to note that this result was obtained on the dataset that we had at that time - due to the specifics of its construction, it contained errors and needed to be cleaned and reformatted.

However, after building a new, cleaned dataset (the training, validation, and test samples were completely rebuilt) with higher data quality, the F1-score value increased to 86.9%. Despite the very significant improvement in results, direct comparison of indicators is complicated by differences in the quality of the datasets used. Nevertheless, it is logical to assume that with an increase in the quality of the training dataset, the efficiency of the model increases, and with an increase in the quality of the test sample, we get more accurate estimates.

## 3 Classic Problems

The model presented earlier in the context of operator automation helps solve practical problems, but classic problems for neural network computer vision remain. One of them is that the felling is found by the model, but the contours are unsatisfactorily filled. As a result, the specialist has to edit the felling contours before sending it to the services. This is easier than searching for fellings without any help from artificial intelligence. However, this requires additional time.

Another problem is that the model completely misses forest fellings in some cases. A person who could notice these fellings on their own will not see them at all, since they are heavily dependent on the hints of the neural network model. Not relying on these hints and rechecking large areas manually is behavior that contradicts automation.

The third classic problem of neural network machine vision is the presence of false positives, the number of which is desirable to minimize. Cases of false positives require additional attention from the operator, which also increases the time spent on data processing. Therefore, reducing the frequency of false positives is a key direction for improving the efficiency of the entire system.

## 4 Domain specific problems

There is another specific problem, characteristic of our subject area. Despite the high efficiency of the SegFormer-B5 model, there remains the issue of recognizing small objects, such as cut roads and small clearings. These types of clearings are determined by the proposed baseline with rather low efficiency.

We attribute this to the fact that the width and height of the output mask of the model in this architecture are actually four times smaller than the input image. It is logical to assume that the prediction density is therefore not large enough to recognize small and thin target objects. To obtain a mask of the original resolution that matches the input image in width and height, bilinear interpolation is applied to it. However, interpolation does not fully solve the problem of insufficient prediction density. This drawback is insignificant for most widely used datasets, where such objects are few. However, our dataset is characterized by the presence of many small objects, and their number is large enough to significantly affect the results.

Based on the above, the aim of this work was to define a more efficient neural network model for forest felling segmentation, with a particular focus on improving the recognition of fine small target fragments, thereby providing an overall improvement in the quality of modeling.

## 5 Dataset

It should be noted that the described task has a high degree of development for the world's largest tropical forest, the Amazon. This is easily explained by the excessive intensity of its deforestation, the natural desire to preserve the ecosystem that affects the entire planet (by 2020, 25% of the Amazon forest area has been lost). However, deforestation in the northern regions of the world is not such a priority research topic, so it is difficult to find large datasets suitable for us.

In addition, the terrain of the Khanty-Mansiysk Autonomous Okrug-Yugra has its own characteristics (sharp seasonal changes, snowiness, swampiness, high cloudiness, etc.). Naturally, such conditions as in the Khanty-Mansiysk Autonomous Okrug - Yugra are not represented in datasets focused on the study of tropical forests. Therefore, direct use of existing data would be inappropriate. In this regard, we developed our own dataset from satellite images of our region.

A total of 2,600 pairs of fragments of Sentinel-2 summer images covering 5.12 x 5.12 km were labeled. A labeled pair is a set of a "before" frame, an "after" frame, and the markings of the fellings that appeared in the "after" frame. Based on these fragments, 58,177 labeled pairs of frames measuring 256 x 256 pixels were formed (this was done using a sliding window with an overlap of 50%), which were used in training. Examples of frames are shown in Figure 1. Then these frames were divided into samples of the following sizes. Training sample – 40,052 pairs. Validation sample – 12,500 pairs. Test sample – 5,625 pairs.

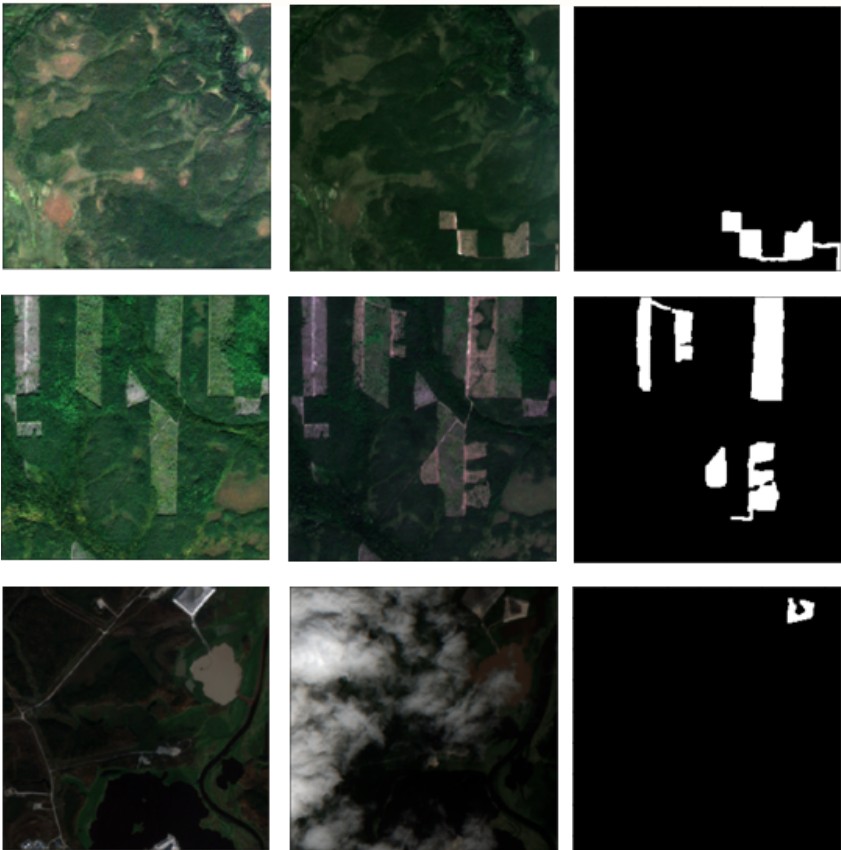

Figure 1: Examples of labeled frames. For clarity of frames, min-max normalization of Sentinel-2 images was used.

# 6 ASSESSMENT AND METRICS

As metrics of the effectiveness of semantic segmentation we use the classic ones: Precision, Recall, F1-score

$$Precision = \frac{TP_{seg}}{TP_{seg} + FP_{seg}} \tag{1}$$

$$Recall = \frac{TP_{seg}}{TP_{seg} + FN_{seg}} \tag{2}$$

$$F_1 = \frac{2 * Precision * Recall}{Precision + Recall} \tag{3}$$

Where $TP_{seg}$ (true positive for segmentation) – number of pixels on the output mask correctly assigned to cuts. $FN_{seg}$ (false negative for segmentation) – number of pixels on the output mask incorrectly assigned to non-cuts. $FP_{seg}$ (false positive for segmentation) – number of pixels on the output mask incorrectly assigned to cuts.

It is worth noting that the previous studies, as well as the current ones, use macro-metrics. This means that the precision, recall and F1-score indicators are first calculated for each test labeled pair separately, and then the average value is calculated for each of them. This approach differs from micro-metrics, where global tp, tn, fp and fn are obtained for the entire dataset, and then the global precision, recall and F1-score values are calculated.

We also evaluate the quality of road recognition. To do this, we take the roads and grab the radius. But for this, a simple road recognition algorithm was needed. An algorithm for selecting roads on a mask was developed. The idea is to perform the erosion operation (available in the opencv library). During this operation, pixels are cut off at the boundaries of figures. Then, we perform the inverse operation - dilation (pixels are added to each figure). The point is that thin fragments (in our case, roads) will not be restored after these steps. It remains to subtract from the original mask what we got above.

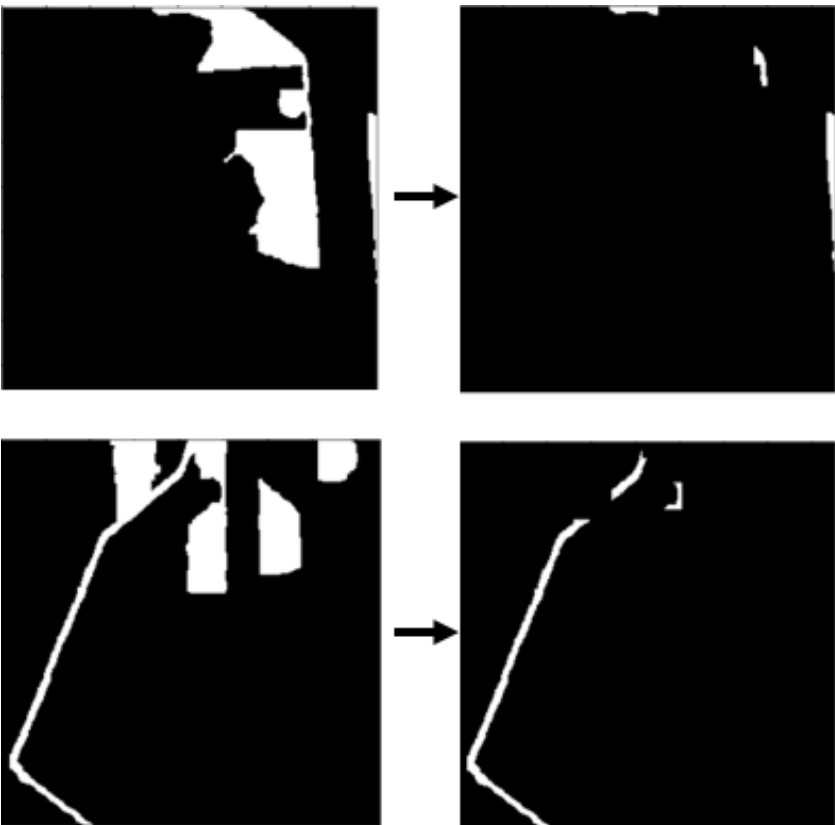

Figure 2: Road segmentation based on morphological operations erode and dilation

## 7 PREPROCESSING AND CORRECTION

To normalize the pixel values of the input image, we divide each pixel by 10,000. For values that exceed this value, we write 10,000 to them to cut off excessively high pixel values. This method

showed better results compared to the common z-normalization. A study was conducted on the capabilities of the transformer neural network segmentation model SegFormer-B5, including the potential for improving the quality of segmentation through the use of atmospheric correction. No noticeable strong positive effect was found, so we decided to abandon atmospheric correction in the current work.

## 8  INPUT DATA FEED

The data feed scheme remained unchanged. For each frame, the following channels of the Sentinel-2 satellite image were used: B2, B3, B4, B5, B11, B12. As for the feed scheme of data pairs as input, images are fed to the neural network as a stack, that is, two images are combined by concatenating their channels. As a result, the number of channels in the new image is the sum of the channels of both original images. In our case, we get 12 channels (6 channels from the "before" frame, the same number from the "after" frame).

## 9  PRELIMINARY DECISION

During the experiments, we came to the conclusion that it makes sense to consider convolutional architectures again, assuming that previously tested approaches may demonstrate different results on cleaned datasets. As a result, to solve problems related to small object recognition, we chose the U-Net++ architecture [5] based on the lightweight MobileNetV2 encoder. Despite the fact that the U-Net++ architecture showed relatively modest results on the "dirty" dataset [2], it demonstrated significantly better results on the cleaned one. If the baseline on SegFormer-B5 gave F1-score = 86.9%, due to Unet++ we managed to get F1-score = 88.7%. In view of the above, we decided to focus on this architecture. Next, we applied mosaic augmentation. In the process of choosing a different number of frames, we came to the following results. As a result, when adding 40 thousand to the training sample described earlier, we obtained the value F1-score = 89.7%.

## 10  ENSEMBLE OF MULTI-SPECIALIZED SEGMENTERS

We select the main model described in the previous section, the task of which is to identify all clearings. In addition to it, a second model is introduced, specializing in road recognition (hereinafter we will call it the "road model"). This model serves as a kind of insurance for the main model. The scheme of this approach is shown in Figure 3. We implement specialization as follows: we obtain forecast masks from both models. We do not take forecasts from the second model directly, but apply a filter to them that preserves only extended objects. The extent is estimated using the elongation index, based on circularity index [6].

$$E = 1 - \frac{4\pi S}{P^2} \tag{4}$$

Where S - the area of the predicted shape, and P is its perimeter.

The closer the shape is to a circle, the closer this index is to zero. The filter is used based on the assumption that when focusing on specific objects (e.g. roads), the model is worse at recognizing other types of objects. Therefore, we leave only extended objects, reducing the probability of erroneous predictions. Then we combine both masks using the logical "OR" operation, forming the final segmentation mask.

With this more correct method, there is an advantage in terms of display. In this table, we see that the overall macro f measure has increased slightly. The fact is that it is taken across the entire dataset. Due to the road segmenter, which is offered in the "Metrics" section, we get a road mask, apply dilate with a value of 10 to get not only roads, but also their outskirts. And we calculate metrics only on these values.

The full assessment is presented in table 1. We can observe a slight increase. In fact, the increase is due to small objects that are poorly reflected by this indicator. For a more visual display, we use a different calculation scheme. With this evaluation (table 2) the results become more visible. The

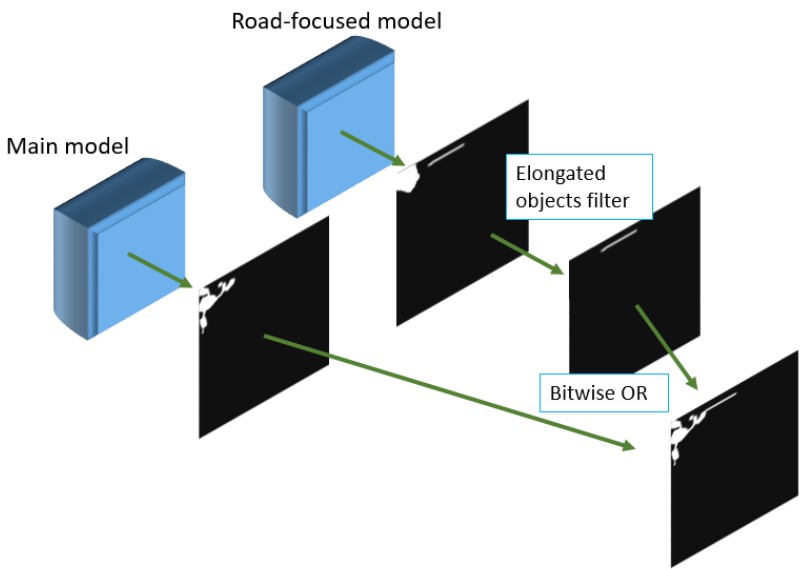

Figure 3: Ensemble of multi-specialized segmenters diagram

Table 1: results of metrics calculation.

| MODEL | F1 score |
|---|---|
| baseline [1] | 0.869430 |
| Unet++ on default dataset | 0.887228 |
| Unet++ on mosaic augmented datased | 0.897139 |
| Ensemble of multi-specialized segmenters | 0.900461 |

road detection is more noticeable - we have an increase of 1.9% compared to Unet++ on mosaic augmented dataset, which confirms that the increase in efficiency by 0.3% in general F-score did not yet characterize the full picture of changes.

## 11 COMPARISON

We present a table (table 3) that shows the results of the assessment for similar/analogous studies. It is important to take into account that comparative analysis is difficult due to the different sample sizes used in the studies presented below, as well as possible nuances in the assessment. Nevertheless, these presented data allow for a general comparative analysis.

## 12 FURTHER ACTIVITIES

Next, we plan to experiment with varying the size of the input window that is fed to the neural network. We assume that by reducing the window, less unnecessary information will be fed to the neural network, and the neural network will be able to focus on more important things. If this assumption is confirmed, this should lead to an increase in the quality of segmentation. There is reason to believe that this step may be justified due to the fact that the size of the felling in the context of the resolution of the satellite images we use (Sentinel-2). It turns out that the average felling can be within a square about 30-40 pixels wide, i.e. we are dealing with low-resolution objects.

Table 2: Road-focused assessment results

| MODEL | F1 score |
|---|---|
| Ensemble of multi-specialized segmenters | 0.872 |
| Unet++ on mosaic augmented datased | 0.853 |

Table 3: Comparison assesment

| Source | F1-score |
|---|---|
| Proposed solution | 0.90 |
| [7] | 0.48 |
| [8] | 0.78 |
| [9] | 0.96 (average over 3 datasets) |
| [10] | 0.88 |

## 13  CONCLUSION

A technique has been proposed that has led to an increase in the efficiency of forest cover disturbance segmentation using a neural network model. The accuracy assessment using the proposed model on the test data set by the F1-score was 90.0%, which is 3.1% higher than the previous solution we proposed. The implemented model can be used to track the state of forests in northern territories, as well as in the broader context of mapping based on data from the Sentinel-2 satellite. The results obtained can be used in other subject areas related to remote sensing of the Earth, neural network segmentation in general.

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
