# OpenReview forum: "Model for segmentation of forest logging in northern latitudes using Sentinel-2 images based on an ensemble of multi-specialized segmenters"
_mathai.club/MathAI/2025/Conference — MathAI 2025 Oral_

### Official Review · Reviewer_q1cc · 2025-02-23
**Review of "Model for Segmentation of Forest Logging in Northern Latitudes Using Sentinel-2 Images Based on an Ensemble of Multi-Specialized Segmenters"**

**Rating:** 7
**Confidence:** 4

**Review:**

Brief Summary. This paper proposes an ensemble approach for detecting illegal logging using Sentinel-2 imagery. By combining a primary segmentation model with a specialized road segmenter, the method improves the detection of forest clearings and roads on a custom dataset from the Khanty-Mansiysk Autonomous Okrug–Yugra region.

Detailed Review. Quality & Clarity
The paper is well-organized and provides a solid description of its methodology, including dataset creation, normalization (dividing pixel values by 10,000), and bilinear interpolation. It compares architectures like U-Net++ with MobileNetV2 and SegFormer B5. A filtering method is introduced with a simple formula:
E = (S / P^2) * 4pi
(where S is area and P is perimeter), though additional explanation would improve clarity.

Originality and Significance. The study addresses a less-explored region (northern forests) compared to tropical deforestation. The ensemble strategy and tailored dataset are innovative, and the method shows an increase in F1 score from 86.9% (baseline) to 90.0%. While the improvement is incremental, it is practically significant for reducing monitoring costs and enhancing detection accuracy.

Comparison with Other Studies. Reference [10] reports an F1 score of 0.96, which is an average over three datasets. This high average may hide variability; one dataset might yield lower performance than the average suggests, and differences in dataset characteristics complicate direct comparisons.

Pros and Cons
Pros: Addresses a real-world, environmentally significant problem.
Develops a robust, region-specific dataset.
Employs an innovative ensemble approach with specialized segmentation.
Cons: Improvements are incremental.
Some methodological details, especially regarding the filtering formula, need clearer explanation.

---

### Official Review · Reviewer_7PFQ · 2025-02-27
**The article describes a U-Net-based approach for determining deforestation**

**Rating:** 6
**Confidence:** 4

**Review:**

The article is devoted to a relevant topic. The subject area and its features affecting the architecture of the solution are well understood. However, there are a number of observations: 1) It is unclear why the F1 measure was chosen. Is there a Precision or Recall domain priority?
2) Table 3 does not specify the methods being compared, which makes it difficult to read the article.
3) Training schedules are not provided. It is difficult to conclude from this that there is no retraining
4) segmentation metrics are not provided, for example, IoU. From this, it is difficult to draw a conclusion about the accuracy of border selection.

---

### Decision · Program_Chairs · 2025-03-08

**Decision:**

Accept (Oral)

**Comment:**

Your article has been accepted and you can make a presentation on the article. All articles will be sorted by rating and within the available conference places one author from each article will be invited. If there are not enough places, then you will either have the opportunity to present remotely or come at your own expense!